

# Impact of model resolution on Holocene climate simulations of the Northern Hemisphere

Axel Wagner[1,2], Gerrit Lohmann[1,2,3], Matthias Prange[2,3]

[1] Alfred Wegener Institute for Polar and Marine Research, Bremerhaven, 27570, Germany
[2] University of Bremen, Bremen, 28359, Germany
[3] MARUM, Center for Marine Environmental Sciences at the University of Bremen, Bremen, 28359, Germany

*Correspondence to*: A. Wagner (post@axelwagner.eu)

**Abstract.** This study demonstrates the dependence of simulated surface air temperatures on variations in grid resolution and
resolution-dependent orography in simulations of the Mid-Holocene. A set of Mid-Holocene sensitivity experiments is carried out with the atmospheric general circulation model ECHAM5 forced with sea surface temperature and sea ice fields from coupled simulations. Each experiment was performed in two resolution modes: low (~3.75°, 19 vertical levels) and high (~1.1°, 31 vertical levels). Results are compared to respective preindustrial runs. It is found that the large-scale temperature anomalies for the Mid-Holocene (compared to the preindustrial) are significantly different in the low- and high-resolution versions. For
boreal winter, differences are related to circulation changes caused by the response to thermal forcing in conjunction with orographic resolution. For summer, shortwave cloud radiative forcing emerges as the predominant factor. In summary, the simulated Mid-Holocene temperature differences (low versus high resolution) reveal a response that regionally exceeds the Mid-Holocene to preindustrial modelled temperature anomalies, and show partly reversed signs across the same geographical regions. Our results imply that climate change simulations sensitively depend on the chosen grid resolutions.

**1 Introduction**

A source of uncertainty in atmospheric climate models is addressed to their limited spatial and temporal resolution (Randall et al., 2007, Jung et al., 2010, Knutti and Sedláček, 2012, Flato et al., 2013, Masato et al., 2013). Small-scale physical processes that cannot be resolved adequately on the computational mesh are represented as parameterised schemes. By improving computer architectures, higher resolved climate models are applied. Model results of simulations with enhanced grid resolution
demonstrate more realistic climate simulations, e. g. due to the growing set of model processes and phenomena (Roeckner et al., 2006, Bader et al., 2008, Reichler and Kim, 2008, Lauer and Hamilton, 2013). Precipitation biases and cloud processes are improved significantly (Roeckner et al., 2006, Doi et al., 2012, Lauer and Hamilton, 2013). For instance, set of CMIP5 model simulations under paleoclimate conditions demonstrate the general underestimation of summer precipitation where the bias can be addressed to cloud physical processes (Harrison et al., 2014). As a resulting feedback mechanism, the large-scale boreal
summer warming is overestimated (Harrison et al., 2014).





In addition to resolution-dependent difficulties in comparing climate model simulations, model inter-comparisons also show large differences in cloud feedback processes and the water cycle (Lauer and Hamilton, 2013). The grid-resolution dependence of regional and global climate dynamics in numerical simulations of the atmosphere is thus of particular interest (e. g. Rind, 1988). The orographic boundary condition, as an example, shows substantial differences between coarsely and

finely resolved simulations. As a response, the meridional vorticity gradient leads to a modified atmospheric Rossby wave propagation (Charney and Eliassen, 1949, Bolin, 1950, Kasahara, 1966, Yoshino, 1981, Cook and Held, 1992), e. g. in response to sea surface temperature forcing (Houghton et al., 1974, Huang, 1978, Chervin et al., 1980, Hoskins and Karoly, 1981, Simmonds and Smith, 1986, Hoskins and Ambrizzi, 1993). Aside from orographic boundary conditions, the tuning of parameterized subgrid-scale processes is indispensable in climate models of varying grid size (Kiehl and Williamson, 1991,

Kristjánsson, 1991, Lane et al., 2000, Tselioudis and Jakob, 2002, Jung and Arakawa, 2004) . In particular, the parameterization schemes have to be developed in order to depict realistic climates for varying model resolutions (Boucher et al., 2013, Flato et al., 2013).

Concerning inter-model comparison studies, experiments have been carried out with a wide spectrum of models involving different parameterization schemes, complexities and resolution (Cubasch et al., 2001, Braconnot et al., 2012, Knutti

and Sedláček, 2012). As one particular example, Roeckner et al. (2006) performed a set of AMIP-style experiments with the atmospheric general circulation model ECHAM5 (Roeckner et al., 2003a). The resolutions ranged from T21L19 to T159L31. While the physics of the model remained unchanged, resolution-sensitive parameters were changed (Roeckner et al., 2006). The results show a tendency to reduce the error upon increasing horizontal and vertical resolution. The variations in total cloud cover between varying model versions are associated with modified parameters in the cloud and convective schemes. These

parameters influence the relative humidity, the precipitation efficiency, and thus the cloud cover as well as the cloud water content. The resolution-dependent differences of these variables lead to changes in surface temperature (Roeckner et al., 2006, Dallmeyer, 2008).

Regarding intra-model comparisons for the modern climate, Cess et al. (1990) present a general circulation model intercomparison with focus on global atmospheric feedback processes. The treatment of cloud processes are identified as

responsible for intermodal differences. Tibaldi et al. (1990), Potter (1995), and Jung et al. (2006) analyse the influence of horizontal resolution of the European Centre of Medium Range Weather Forecast (ECMWF) numerical weather prediction model on systematic errors, cloud radiative forcing, and extratropical cyclone characteristics. Low resolved model simulations do not capture the correct nonlinear dynamics of the extratropics. Also the key characteristics of extratropical cyclones, that are highly sensitive to model resolution, show a tendency to more realistic patterns with increased resolution. Furthermore,

cloud radiative forcing characteristics changes with model resolution. Williamson et al. (1995) conduct climate sensitivity studies with the National Center for Atmospheric Research Community Climate Model (CCM2). Their findings show large differences between low and medium resolution model versions. They conclude, that high resolution models are required to better capture nonlinear processes of medium scales. Pope and Stratton (2002) use the Hadley Centre climate model, HadAM3 (Hadley Centre Atmospheric climate Model version 3, the climate version of the Met Office's Unified Model) to perform



resolution dependent sensitivity studies. Model biases are reduced with increased model resolution. Gao et al. (2006) perform simulations with the Regional Climate Model (RegCM2). The sensitivity studies focus on changes in orography and model resolution with focus on East Asia precipitation. The high resolution simulation (< 60 km grid box) with coarse orography provides more realistic results compared to the coarse resolution model and orography. Boville (1991) explores significant improvements in simulating the troposphere towards increased model resolutions. Hack et al. (2006) and Gent et al. (2010)

perform climate sensitivity experiments with the Community Climate System Model (CCSM) Community Atmosphere Model version 3 (CAM3). High resolution simulations show a robust systematic enhancement of atmospheric processes. Furthermore, sea surface temperature and precipitation biases are reduced with model resolution. Hamilton (2006) reviews numerical model simulations with varying resolutions. Increased resolution lead to significantly overall improved global-scale circulations.

For paleoclimate conditions, experiments have been standardized and compiled by the Paleoclimate Modelling

Intercomparison Project (Gladstone, 2005, Braconnot et al., 2007a, Braconnot et al., 2007b, Brewer et al., 2007, Zheng et al., 2008, Otto-Bliesner et al., 2009, Braconnot et al., 2012, Lohmann et al., 2013). Here, we supplement such intercomparison studies concerning potentially resolution-dependent results. In contrast, our experimental setup is restricted here to one single model, but different resolutions under fixed boundary conditions in SST and sea ice concentration.

The Last Glacial Maximum (LGM) has been examined, for instance, using the U.K. Universities Global Atmospheric

Modeling Programme (UGAMP) general circulation model (GCM) (Dong and Valdes, 2000). Systematic differences between low and medium resolution simulations are shown. Low model versions struggle to simulate planetary waves and storm tracks. Furthermore, high resolution model simulations better match with geological findings. Jost et al. (2005) analyse model simulations with proxy datasets. The low resolved simulations show a large discrepancy towards pollen-based paleoclimate reconstructions. High-resolution simulations show no reduction of this discrepancy, however are able to detect more realistic

precipitation patterns. Kim et al. (2008) compare low and high resolved LGM simulations. Surface temperature and precipitation patterns of the high resolved simulation are in better agreement with proxy estimates compared to the low resolved simulation. Vavrus et al. (2011) focus on the role of model resolution in simulating glacial inception. Atmospheric pressure, onshore moisture fluxes and thus snowfall over land differ significantly between the employed model resolutions.

To our knowledge, no intra-model study has been performed for the Holocene epoch so far, even though the Mid-

Holocene (6 ka BP) is one of the most frequently simulated time slices in paleoclimate modelling and belongs to the standard PMIP experiments (e. g. Braconnot et al., 2007a, Braconnot et al., 2007b, Braconnot et al., 2012, Lohmann et al., 2013). Compared to the Last Glacial Maximum, temperature anomalies of the Mid-Holocene are much smaller, such that the ratio between model biases and simulated temperature signals is expected to be relatively large (Hargreaves et al., 2013, Lohmann et al., 2013).

In this study we examine the impact of model resolution on Holocene surface temperature variations and focus on the high northern latitudes. Chapter 2 briefly describes the setup of the performed model simulations. Chapter 3 depicts the most prominent meteorological differences of the resolution and orography dependent sensitivity studies. Chapter 4 discusses the results over other studies focusing on over climate periods as the last glacial maximum and the modern era. Chapter 5



summarizes the major findings, draws conclusions and points out the difficulties in comparing model simulations of unequal
resolution.

## 2 Methods

Here, we employ the atmosphere model ECHAM5 (Roeckner et al., 2003a) in a stand-alone mode to isolate resolution effects
on atmospheric processes in Mid-Holocene simulations. The model has been tested in various resolutions by Roeckner et al.
(2003b) and Roeckner et al. (2006) against observational datasets. Except for resolution-dependent parameter changes, the
model physics remains identical. Some resolution-dependent parameters are related to the horizontal diffusion, the orographic
drag scheme or the adjustment time scale in the convective parameterization. As reported earlier, other parameters are tuned
to match the global mean radiation balance at the top of the atmosphere (Roeckner et al., 2006).

Six experiments are presented here (Table 1), comprising the Mid-Holocene (MH) and preindustrial (PI) periods.
Runs were performed in two resolution modes: low (horizontal: ~3.8°, vertical: 19 levels) and high (horizontal: ~1.1°, vertical:
levels), referred to as $LR_{MH-PI}$ and $HR_{MH-PI}$, respectively. Furthermore, computer parallelization schemes, code and compiler
structure were identical. Greenhouse gas concentrations were adjusted to the appropriate conditions following the PMIP2
convention (Flückiger et al., 1999, Monnin et al., 2001). The varying orbital forcing parameters were computed following
Berger (1978). For simplicity, land surface conditions, aerosols and ozone were set to present-day conditions.

The performed sensitivity experiments ($LR_{MH-PI}$, $HR_{MH-PI}$ and $HR_{MH-PI}(LR_{oro})$) are branched out into a MH and a PI
time-slice run. For all six experiments an integration time of 50 years (i. e. MH) was used, where the first 10 years (i. e. 100
orbital years) were regarded as the spin-up phase and excluded from further analysis. Experiments $LR_{MH-PI}$ and $HR_{MH-PI}$ capture
the isolated effect of model resolution on the climate simulations. In the framework of $HR_{MH-PI}(LR_{oro})$, the finely resolved
$HR_{MH-PI}$ orography was replaced by a $LR_{MH-PI}$ orography in order to isolate the effect of orographic resolution on the climate
simulation (Fig. 2). SST and SI fields are prescribed (Fig. 1). They were derived from transient model simulations (Lorenz and
Lohmann, 2004) performed with the coupled general circulation model ECHO-G (Legutke and Voss, 1999). The surface
boundary conditions are used as anomalies from the reanalysis datasets provided by AMIP2 (Taylor et al., 2000). This
technique results in realistic variability in SST and SI model forcing fields. Setting up the high-resolution ECHAM5
simulations, a bilinear interpolation of SST and SI (from 3.8° to 1.1°) is applied (cf. Herold and Lohmann, 2009).

For our analysis, the geographical focus is on the northern extratropics (north of 40°N). Results are presented as
anomaly (MH minus PI) and anomaly difference plots (e. g. $LR_{MH-PI}$ anomaly minus $HR_{MH-PI}$ anomaly) where we focus on the
seasonal analysis of simulated temperatures. The anomalies of the variables are calculated as the mean state difference of the
MH minus PI period. Further model simulations cover sensitivity analysis of low and high-resolution simulations as well as
isolated effects of orographic changes.





## 3 Results

### 3.1 Temperature differences during winter

The boreal winter temperatures two metres above ground (T2m) MH minus PI anomaly in $LR_{MH-PI}$ (Fig. 3a) ranges from -2.7 to 2.0 K with the largest positive anomalies across Eurasia and cold anomaly spots in the areas of the Labrador and Nordic Seas (Fig. 3a). The high-resolution experiment $HR_{MH-PI}$ shows a consistent T2m pattern across the oceanic areas, however with more pronounced cold spots of up to -4.5 K (Fig. 3b). Across Eurasia, the $HR_{MH-PI}$ T2m anomaly pattern falls into two parts

with a maximum Holocene anomaly of 2.2 K in northern Siberia. In North America, notably across the Rocky Mountains Range and leeward, the $HR_{MH-PI}$ experiment shows a significant increase of the T2m anomaly of 3.2 K (Fig. 3b).

     The isolated effect of grid resolution and parameterization adjustments arises from the difference of the low-resolution experiment $LR_{MH-PI}$ and the high resolution experiment with a low resolved orographic mask $HR_{MH-PI}(LR_{oro})$. The boreal winter (December-January-February, DJF) T2m anomaly differences (Fig. 3d) range from -1.25 to 3.0 K with the greatest differences

across Eurasia, eastern North America as well as across the Labrador Sea. In summary, the DJF T2m anomaly differences (Fig. 3d) across large areas of Eurasia and North America are of the same order of magnitude as the Holocene T2m anomalies (Fig. 3a, b). The T2m anomaly differences during DJF (Fig. 3d) are mostly a consequence of changes in atmospheric circulation (Fig. 4). A detailed description follows in sect. 3.4.

### 3.2 Temperature differences during summer

During boreal summer, the Holocene $LR_{MH-PI}$ T2m anomalies (Fig. 5a) are intensified compared to $HR_{MH-PI}$ (Fig. 5b) and $HR_{MH-PI}(LR_{oro})$ (Fig. 5c) across western and eastern Eurasia and weakened across North America. However, the basic patterns and magnitudes ($LR_{MH-PI}$: -0.8 to 2.6 K; $HR_{MH-PI}$: -1.0 to 2.9 K) of T2m anomalies are consistent (Fig. 5a, c).

     The experiments result in less pronounced summer (June-July-August; JJA) changes (Fig. 5d) compared to DJF (Fig. 3d). During JJA, the large-scale anomaly differences mainly remain in the range of ± 1 K (Fig. 5d). The resolution-induced

anomaly differences of summer T2m are in some regions still of the same order as the Holocene T2m anomalies (Fig. 5a, b). The Holocene temperature anomaly differences (Fig. 5d, f) are influenced by variations in shortwave cloud radiative forcing, which are in turn affected by alterations of the total cloud cover (Fig. 6a, c). Total cloud cover anomaly differences between $LR_{MH-PI}$ and $HR_{MH-PI}$ (Fig. 6a) show relative changes of ± 10 %. Differences in shortwave cloud radiative forcing (Fig. 6b) lead to JJA variations of -22 to 24 W/m² across Eurasia and North America. For JJA, the spatial difference patterns of the

shortwave cloud forcing (Fig. 6b) and T2m (Fig. 5f) are quasi-coherent and, thus, shortwave cloud forcing can be regarded as the major driving factor of the summer T2m differences.

### 3.3 The isolated effect of orography

The low T31L19 (Fig. 2a) and high T106L31 (Fig. 2b) resolved orography displays similar maximum heights of mountain ranges across the Qinghai-Tibetan (Qingzang) Plateau (5201/5117 m), Greenland (3086/3055 m) and the Transantarctic



Mountains (3973/3777 m). The geographic locations of these peaks and chains (T31 versus T106) are shifted up to several hundreds of kilometers. The shifts are dependent on the orographic resolution. Across the North (2718/2195 m) and South American Cordillera (4030/3684 m), differences in altitude amount to several hundreds of meters. In total, resolution-dependent differences in altitude (Fig. 2) show an absolute spread of ± 1600 m for the South American Cordillera, ± 1300 m for the Qinghai-Tibetan (Qingzang) Plateau (due to the shift in the geolocation), ± 900 m for Greenland and Antarctica  and ±

700 m for the North American Cordillera.

The coarse orographic mask, as applied for the HR$_{MH-PI}$ (LR$_{oro}$) experiment, leads to less pronounced DJF T2m anomaly patterns across Eurasia and North America (Fig. 3c) compared to HR$_{MH-PI}$ (Fig. 3b). For the HR$_{MH-PI}$ (LR$_{oro}$) experiment, regional T2m pattern range from -2.0 to +1.5 K (Fig. 3c), whereas, for the HR$_{MH-PI}$ experiment, values cover the range from -1.0 to + 5.0 K.

The comparison of HR$_{MH-PI}$ and HR$_{MH-PI}$ (LR$_{oro}$) highlights the orographic sensitivity of T2m anomalies during the Holocene (Fig. 3e). For instance, across the Rocky Mountain Range, T2m anomalies (HR$_{MH-PI}$, Fig. 3b) display a pronounced and significant amplitude of up to +5.0 K. The replacement of the orographic mask (T106 against T31) results in a smoothed temperature signal characterized by a change of only +1.5 K (Fig. 3c).  The isolated effect of orography is displayed as T2m anomaly differences (Fig. 3e), amounting to +3.0 K across northern Eurasia and western and central North America (Fig. 3e).

As a response to the different orography (Fig. 3c), the sea-level pressure distribution of HR$_{MH-PI}$(LR$_{oro}$) remains conserved (Fig. 4c) compared to HR$_{MH-PI}$ (Fig. 4b), albeit partly reinforced and twisted to the west (approximately 30 to 45°). Exceptions are the low- and high-pressure cells over North America and Greenland with reversed pressure anomalies (Fig. 4b and c). The geopotential height field (Fig. 4f) mirrors a strong barotropic response across the Northern Hemisphere, north of 40°N. The stationary wave pattern (Fig. 4e) is transformed substantially (Fig. 4f). Warm air masses are advected onto the

continents of North America and western Eurasia, inducing positive land temperature anomalies.

In total, the orographic effect (Fig. 3e) cannot explain all the DJF T2m differences. Only the combined effect of resolution and orographic changes explains the T2m anomaly differences (Fig. 3f). The T2m differences across eastern North America and several regions of Eurasia exceed the orographic-induced T2m differences.

During boreal summer (Fig. 5c), large-scale patterns of T2m anomalies are of the same order as HR$_{MH-PI}$ (Fig. 5b)

and similar across continental Eurasia and North America. In contrast to DJF (Fig. 3e), anomaly differences during JJA (Fig. 5e) are less pronounced than the Holocene T2m anomalies (Fig. 5a and b). The isolated effect of a replaced orographic mask (finely versus coarsely resolved mask) shows large-scale T2m differences in the range of ± 1 K (Fig. 5e). It is found that these T2m differences can be accounted for by variations in cloud distribution, thus affecting the shortwave cloud radiative forcing balance. However, the resolution-induced T2m differences (LR$_{MH-PI}$ - HR$_{MH-PI}$; Fig. 5d) cannot be attributed sufficiently to the

changed orography (Fig. 5e).



### 3.4 The combined effect of orography and resolution

With our experiments, we can evaluate the resolution effect of the full dynamics (Fig. 3d and 5d), the effect due to orography only (Fig. 3e and 5e), and the combined effect (Fig. 3f and 5f) on T2m. The isolated effects of orographic resolution and dynamics are identified for both boreal summer and winter. During DJF, the effects are amplified by the factor three (Figs. 3d, e; 4d and e).

The DJF T2m anomaly differences ($LR_{MH-PI}$ minus $HR_{MH-PI}$; Fig. 3f) range from -2.5 to 2.0 K with the greatest differences across the Rocky Mountains Range, farther east in North America as well as across central and northeastern Eurasia. In summary, the DJF T2m anomaly differences (Fig. 3f) across large areas of Eurasia and North America are of the same order of magnitude as the Holocene T2m anomalies (Fig. 3a, b). The T2m anomaly differences during DJF (Fig. 3f) are mostly a consequence of changes in atmospheric circulation. In the low-resolution simulation ($LR_{MH-PI}$), the mean sea level pressure (Fig. 4a) and the geopotential height pattern (Fig. 4b) in the middle atmosphere display a quasi-annular barotropic structure in the circulation anomaly (Fig. 4a, d). In the high-resolved version ($HR_{MH-PI}$), a more complex barotropic wave-like pattern with higher wavenumber emerges (Fig. 4b, e). As a result of the perturbed zonal air flow, the dynamically developed troughs and ridges affect the meridional and zonal circulation. In the $LR_{MH-PI}$ simulation, across Eurasia a zonal flow anomaly carries warm air onto the continent. As a response, temperatures across central Eurasia rise (Fig. 3a). Further to the east, a high-pressure anomaly in the $HR_{MH-PI}$ experiment transports cold air masses across northeastern Eurasia, causing a decrease in continental temperatures (Fig. 3b). A pronounced low-pressure anomaly over northern North America (Fig. 4b) advects air from the East Pacific Ocean to central North America, creating a positive T2m anomaly in the $HR_{MH-PI}$ run (Fig. 3b). A lowering in surface albedo (max. 10 %) due to snow cover reduction amplifies the T2m changes (not shown). The JJA T2m anomaly differences ($LR_{MH-PI}$ minus $HR_{MH-PI}$; Fig. 5f) are characterized by less pronounced JJA changes compared to DJF (Fig. 3f). During JJA, the large-scale anomaly differences remain in the range of ± 1 K (Fig. 5f). The resolution-induced anomaly differences of summer T2m are in some regions still of the same order as the examined Holocene T2m anomalies (Fig. 5a, b).

### 4 Discussion

A comparison of our results to inter-comparison model studies of the Holocene is taken into consideration. Dimri (2004) and Giorgi and Marinucci (1996) point out the sensitivity of physical parameterizations in climate models as function of their horizontal resolution. For ECHAM5, the subgrid scale orography and the cloud scheme are dependent on the represented model resolution (Roeckner et al., 2003a).

Resolution dependent summer temperature anomalies were analysed by Harrison et al. (2014) A set of CMIP5 model simulations under paleoclimate conditions has been compared. The models generally underestimate summer precipitation and overestimate summer warming (large scale patterns). The effect is addressed to model specific biases in land-atmospheric heat fluxes. The findings of Harrison et al. (2014) are evident in ECHAM5 simulations. The effect has been depicted by Kiehl and

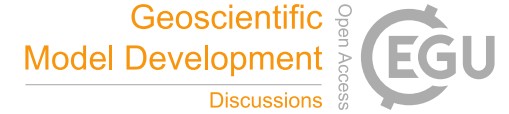

Williamson (1991) and Williamson et al. (1995). The statistical cloud scheme by Tompkins (2002) in ECHAM5 leads to decreases of total cloud cover patterns in ECHAM5 with increased model resolution. The findings are consistent with our mid-

Holocene and preindustrial time-slice experiments. Here, the coarse-resolved simulations show higher total cloud cover distribution over Eurasia and North America compared to the high-resolved simulations. Annual mean differences exceed 15 % on a supra-regional scale, seasonal differences surpass 35 %. The cloud cover differences in turn affect the cloud radiation balance. Nam and Quaas (2012) describe this phenomenon while evaluating ECHAM5 simulations by satellite datasets. They demonstrate that ECHAM underestimates cloud radiative forcing when total cloud coverage is large. The downstream process

influences the evaporation and hence the water cycle in the model as well.

Our summer temperature distribution across Eurasia and North America is in agreement with earlier results published in PMIP2 and PMIP3 (Braconnot et al., 2007a, Braconnot et al., 2012, Lohmann et al., 2013). These simulations (ensemble median) capture a Holocene cooling trend of summer surface temperatures over continental Eurasia and North America with a maximum of 2 K. Increases in total cloud cover were found to contribute to these changes (Braconnot et al., 2007b, Braconnot

et al., 2012). On the regional scale, T2m anomaly differences of $LR_{MH-PI}$ minus $HR_{MH-PI}$ are above the summer differences between the Mid-Holocene and pre-industrial (Fig. 5f). Bjerknes (1969), Rowntree (1972), and Chervin et al. (1980) discuss the linkage between sea surface temperatures and their influence on atmospheric pressure cells across the tropics and extratropics. As for all performed simulations in this study, sea surface temperature fields remained fixed and thus, are not interactively calculated and do not contribute to changes in the observed atmospheric dynamics. The advantage of fixed SSTs

constitutes at the same time as a disadvantage, since the SSTs did not develop freely and are not completely consistent with the atmospheric circulation pattern, because the circulation would affect the SSTs. Thus, atmospheric feedback processes could be distorted.

For DJF, the combined effect of resolution and orography ($LR_{MH-PI}$ minus $HR_{MH-PI}$) is more pronounced (Fig. 3f) and thus surpasses Holocene temperature changes throughout large areas of continental Eurasia and North America. The sensitivity

study $HR_{MH-PI}$ ($LR_{oro}$) represents the isolated effect of orographic-induced T2m anomalies. Pronounced T2m anomaly differences during DJF are reflected across northern Eurasia and western and central North America (Fig. 3e). The results are in line with the findings of Charney and Eliassen (1949), Grose and Hoskins (1979), and Hoskins and Karoly (1981). They argue that mountain barriers of the size of the Rocky Mountains and the Himalayas influence the position of the stationary Rossby wave. They observed that certain characteristics of the flow pattern in the upper part of the troposphere of the northern

hemisphere during the yearly cycle remained constant, despite the thermal contrast between the continents and the oceans. Kasahara (1966), Kasahara et al. (1973), and Manabe and Terpstra (1974) applied atmospheric general circulation models to isolate the effect of mountain ranges and planetary waves. They ran a set of simulations with and without the effect of mountains and focused on the stationary and transient disturbance of atmospheric wave trains. The presence of mountain barriers led to an increase in the stationary eddy conversion of potential to kinetic energy while the absence of mountains

causes the opposite. Furthermore, Manabe et al. (1970) reported in an earlier study that the coarseness of the orographic mask and the model resolution do play an important role to this phenomenon.



As a response to orographic changes ($HR_{MH-PI}(LR_{oro})$), the formation and distribution of low- and high-pressure cells is influenced (Fig. 4c and f compared to Fig. 4b and d). The redistribution of pressure cells, in turn, affects the T2m anomalies (Fig. 3c compared to Fig. 3b). However, the replacement of the orographic mask (low- versus high-resolved one) cannot explain the whole DJF T2m differences caused by $LR_{MH-PI}$ minus $HR_{MH-PI}(LR_{oro})$ (Fig. 3d). For instance, the central Eurasian T2m anomaly in Fig. 3d is not visible in Fig. 3e. This T2m feature can be attributed to changes in the treatment of the processes on different scales (Fig. 3d). However, the local cloud shortwave radiative forcing (crfsw) on surface-level does not serve as an explanation, due to the influence of the polar night. The changed clouds could affect the energy balance of the subtropical and tropical atmosphere. Furthermore, our result can be linked to Bjerknes (1969), Rowntree (1972), Hoerling et al. (2001), Rimbu et al. (2004) or Li and Lau (2012) discussing a relation between tropical sea surface temperatures and teleconnection patters as the Pacific–North American (PNA) or North Atlantic Oscillation (NAO). Hoerling et al. (2001) state that changes in tropical sea-surface temperatures affect tropical precipitation and latent heating. As a result, the atmospheric circulation in high northern latitudes and, thus, the NAO pattern is influenced. Rimbu et al. (2004) analysed observational and model data in terms of teleconnection patterns between ENSO and NAO. Their analysis shows a seasonal teleconnection dependence: NAO- phases occur more frequently during late winter El Niño events. NAO+ phases occur more frequently during La Niña events. Li and Lau (2012) support these findings by analysing model and observational data.

Orographic effects during JJA (Fig. 5c) are of minor influence to T2m anomaly changes (Fig. 5e). The effects of orographic resolution and clouds are identified for both boreal summer and winter, whereby during DJF the effects are amplified by a factor of three (Figs. 3f and 5f). Furthermore, we found that snow cover and surface albedo across the Rocky Mountain Range and central North America act as an amplifying mechanism to temperature changes.

The isolated effect of orography ($HR_{MH-PI}(LR_{oro})$) shows temperature anomalies during boreal winter that are seen in the immediate vicinity of large-scale mountain ranges. However, parts of the North-Eurasian lowlands are affected as well. The combined effects of $LR_{MH-PI}$ minus $HR_{MH-PI}$ and $HR_{MH-PI}(LR_{oro})$ result in resolution-induced temperature anomalies that are regionally as large as the Holocene temperature anomalies. In particular, $HR_{MH-PI}(LR_{oro})$ shows the effect of the low resolution of orography cannot explain the differences in the $LR_{MH-PI}$ and $HR_{MH-PI}$ Mid-Holocene simulations. The interaction between orography and thermal forcing leads to characteristic stationary wave patterns.

Comparing our DJF temperature anomalies ($LR_{MH-PI}$ versus $HR_{MH-PI}$, Fig. 3a, b) with proxy data (e. g. Bartlein et al., 2011), we see not a clear improvement of simulating some heterogeneities in the $LR_{MH-PI}$ relative to the $HR_{MH-PI}$ experiment. For instance, proxy data suggest a reduced latitudinal temperature gradient over Europe during the Mid-Holocene (Bartlein et al., 2011) which is more pronounced in $HR_{MH-PI}$ than in $LR_{MH-PI}$ (Fig. 3a, b). Across West Siberia proxy and model data (low- and high-resolved versions) show similar positive Holocene temperature anomalies. Across Central Eurasia, the low-resolution model version better reflects the positive Holocene temperature anomalies, compared with the negative T2m anomaly of the higher resolved version. The T2m anomaly differences between the model runs are strongest over Siberia and the Rocky Mountain Range (Fig. 3f). In the area of the Rocky Mountain Range, pollen data and model runs show contrasting (pollen negative, model positive) anomaly signs.





Focusing on JJA T2m anomalies (Fig. 5), low- and high-resolution model runs show continental-wide positive temperature anomalies during the Mid-Holocene. Compared to proxy data, Europe and northern Siberia can be suggested as being consistent with higher Mid-Holocene T2m anomalies (compared to PI, Fig. 5a-b). The results are supported by multi-proxy reconstructions by Sundqvist et al. (2010) who analysed temperature proxies of the northern high-latitudes. The findings
are confirmed by pollen and plant macrofossil records of Prentice et al. (1996), Tarasov et al. (1998), MacDonald et al. (2000), and Seppä and Birks (2001). They stated a southward retreat of the Arctic treeline and concluded a decline in Arctic mid- to late Holocene summer temperatures. Lake records from western Greenland estimate local mid-Holocene summer temperatures of approximately 2 to 3 degrees warmer than present (Axford et al., 2013). $\delta^{18}$O records in ice cores from the Agassiz Ice Cap on Ellesmere Island imply a summer cooling of 4 degrees from 8 ka BP to present (Fisher et al., 1995). A review article by
Briner et al. (2016) reports temperature proxies of Arctic Canada and Greenland. Temperatures decreased by approximately 3 ± 1 °C from the mid- to late Holocene. However, a detailed data-model comparison is beyond the scope of the present paper.

## 5 Conclusions

The resolution dependence of model results has been addressed for present conditions (e. g. Jost et al., 2005, Hack et al., 2006, Roeckner et al., 2006, Byrkjedal et al., 2008, Kim et al., 2008). The studies point out the benefits of higher resolved model
experiments. Our Mid-Holocene climate simulations demonstrate the dependence of continental surface air temperatures on spatial grid resolution and orography changes. The two resolution modes (LR$_{MH-PI}$ and HR$_{MH-PI}$) reveal large-scale temperature anomaly differences that are of the same order of amplitude as the Holocene temperature anomalies (Mid-Holocene minus preindustrial). The differences in boreal winter temperature anomalies are mainly attributable to distinct stationary wave patterns. We suspect that stationary and transient eddies and the model's climatological basic states determine the atmospheric
response to Holocene thermal forcing anomalies, in a similar way as described in other contexts (e. g. Chervin et al., 1980, Held et al., 2002, Kushnir et al., 2002). However, a systematic analysis of the atmospheric dynamics mechanisms is beyond the scope of the present study. Summer temperature differences between the low- and high-resolution models can be attributed to changes in shortwave cloud radiative forcing.

Our results imply that differences in paleoclimate simulations could partly be attributed to the use of different grid
resolutions, even when using the same atmospheric circulation model, but fixed SSTs and sea ice distribution. In future studies, systematic sensitivity experiments shall be performed in order to disentangle the causes of the different wave pattern responses for past, present and future climate change scenarios. It might be that high-resolution models are required to represent the atmospheric dynamics such as baroclinic waves and even blocking phenomena (e. g. D'Andrea et al., 1996, Matsueda et al., 2009, Scaife et al., 2010, Berckmans et al., 2013, Dunn-Sigouin and Son, 2013, Rimbu et al., 2014). Blockings play a central
role for the extratropical circulation, mean climate and extremes (Masato et al., 2013, Ionita et al., 2016).

As a logical next step, one can perform a systematic validation of model simulations based on proxy datasets. In some regions the high-resolution simulation fits better to proxy datasets than the low resolution one, but this does not seem to be the



case everywhere. For palaeomodel intercomparison studies such as PMIP (Braconnot et al., 2007a, Braconnot et al., 2012, Lohmann et al., 2013), we suggest future studies to systematically explore resolution-dependent results in coupled and
uncoupled general circulation models.

**Code and data availability**

Model code is license restricted. Data will be provided upon request via mail (post@axelwagner.eu).

**Acknowledgements**

This research work has been supported by the German Research Foundation (DFG), project ID: LO 895/13-1, PR 1050/3-1.
Model integrations were carried out with facilities provided by the Alfred Wegener Institute for Polar and Marine Research.



**Tables**

Table 1: Overview of experiments

| Experiment | | Time slices | Model grid (resolution) | Orography (resolution) |
|---|---|---|---|---|
| $LR_{MH\text{-}PI}$ | $LR_{MH}$ | MH | T31L19 | T31L19 |
| | $LR_{PI}$ | PI | " | " |
| $HR_{MH\text{-}PI}$ | $HR_{MH}$ | MH | T106L31 | T106L31 |
| | $HR_{PI}$ | PI | " | " |
| $HR_{MH\text{-}PI}(LR_{oro})$ | $HR_{MH}(LR_{oro})$ | MH | T106L31 | T31L19 |
| | $HR_{PI}(LR_{oro})$ | PI | " | " |








**Figures**

Figure 1:

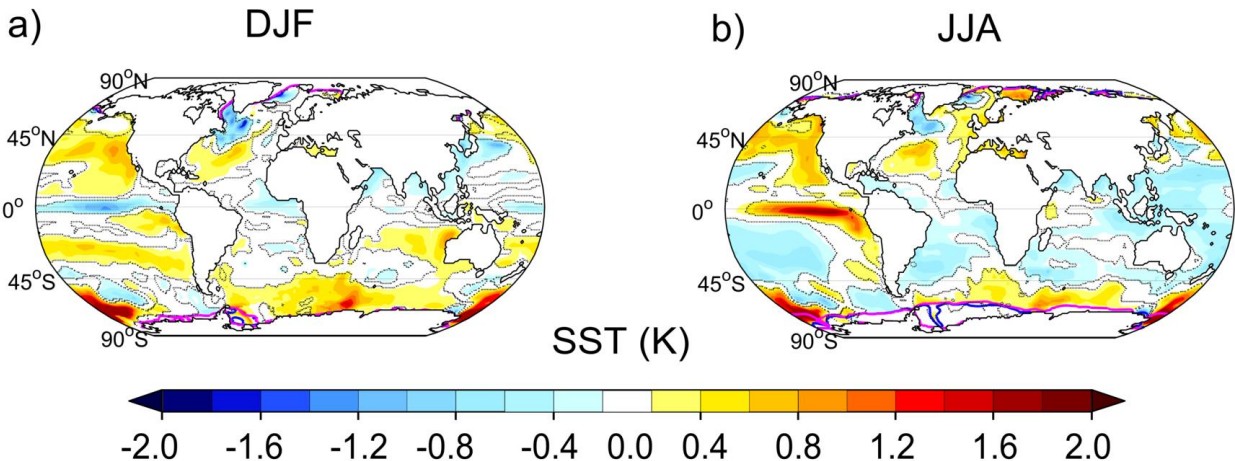




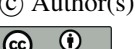


Figure 2:

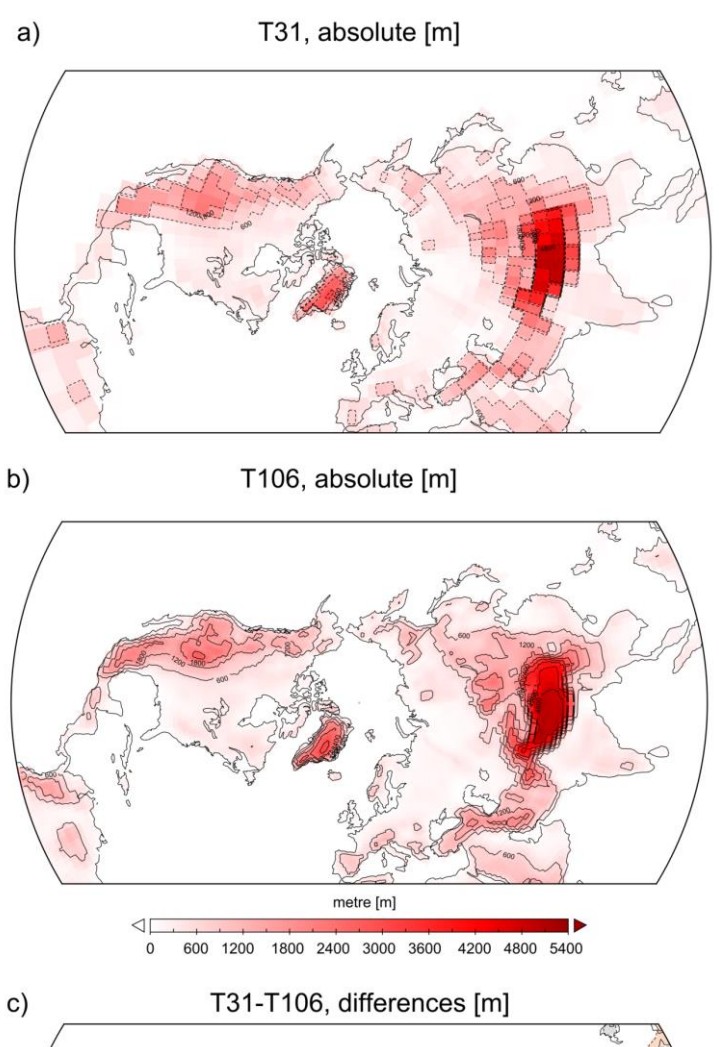

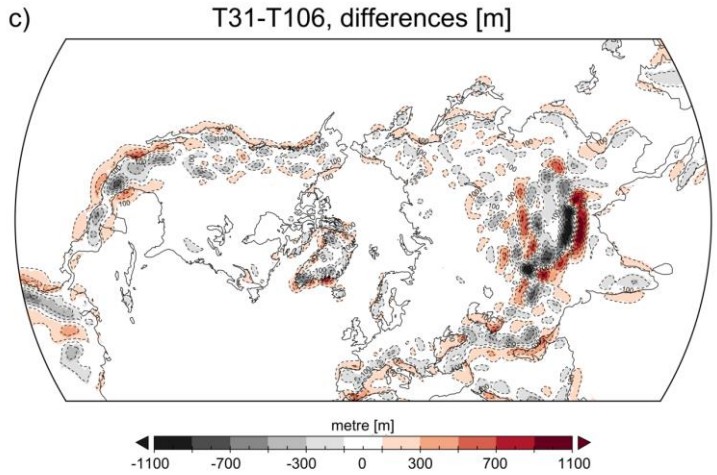





Figure 3:







Figure 4:






Figure 5:



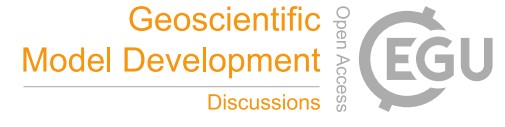

Figure 6:

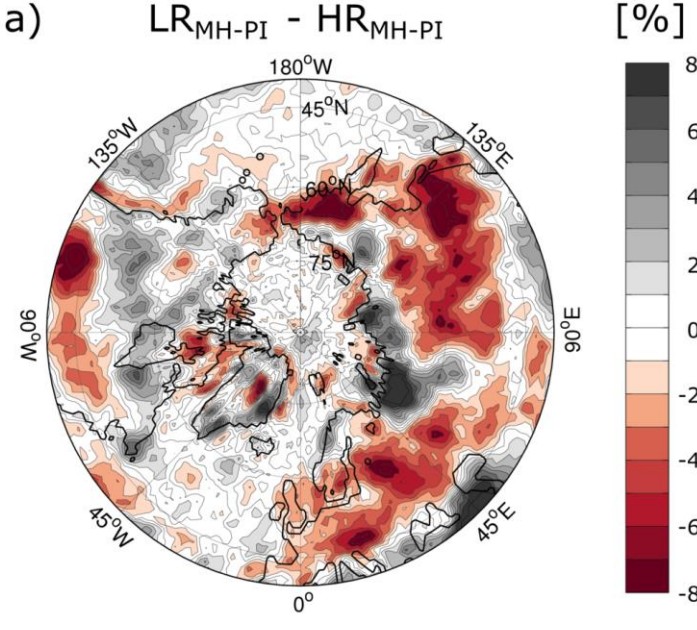

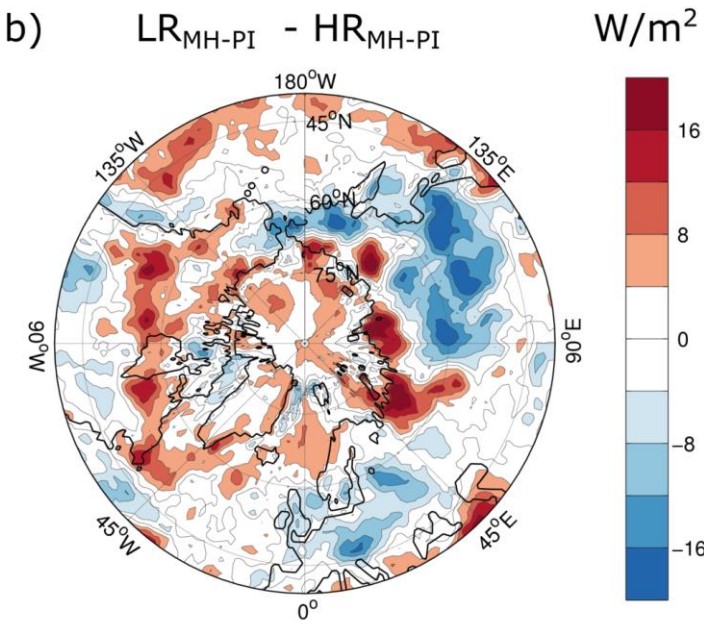




**Figure captions**

Figure 1: Sea surface temperatures (SST) and sea-ice concentrations (SI) for Mid-Holocene (MH) anomalies compared to preindustrial based on transient ECHO-G simulations of Lorenz and Lohmann, 2004. a) DJF anomaly of SST and SI. b) as a), but JJA anomalies. Units are Kelvin [K]. The sea-ice 50% concentration is marked as solid magenta (MH) and dark blue (PI) lines. Stippled lines indicate statistical significance.

Figure 2: Orographic height used for the performed sensitivity experiments. The shown map section covers the largest part of the Northern Hemisphere. The absolute height of the low and high-resolution model orography are shown in a and b). Height anomalies between low- (T31L19) and high-resolved (T106L31) experiments are demonstrated in c). Units are meters [m].

Figure 3: Anomalies and anomaly differences of simulated 2m air temperatures (T2m) for boreal winter (DJF). The area is restricted to the Northern Hemisphere (40–90° N). T2m anomalies are calculated as Mid-Holocene (MH) minus preindustrial (PI) differences and are simulated by two distinct model resolutions (T31L19, T106L19). Low- and high-resolution T2m anomalies are depicted in a) and b). The performance of a high-resolved simulation with a low-resolved orographic mask is shown in c). The subtraction of a) minus b) results in f), the resolution-induced MH anomaly difference of T2m (T31L19[MH-PI]) minus (T106L31[MH-PI]). The subtraction of b) minus c) results in e), the orographic-induced T2m anomaly differences. The subtraction of a) minus c) results in d), the isolated effect of high-resolution dynamics on T2m anomaly differences. Units are Kelvin [K]. Significant areas (alpha = 0.05) are surrounded by dotted black lines. The respective coastlines are depicted by heavy solid black lines.

Figure 4: Sea level pressure and geopotential height at 500 hPa (MH minus PI) anomalies shown for DJF. Sea level pressure anomalies are shown for the low-resolved (T31L19) a) and the high-resolved (T106L31) experiment b). The performance of a high-resolved simulation with a low-resolved orographic mask shows the sea level pressure anomalies in c). Geopotential height anomalies are shown for T31L19 in d), for T106L31 in e) and for the changed orographic mask in f). Units of sea level pressure are in hectopascal [hPa] and for geopotential height in meters [m]. Stippled lines as in Figure 3.

Figure 5: As Figure 3, but for boreal summer (JJA).

Figure 6: Anomaly differences of total cloud cover (cf) and shortwave cloud radiative forcing (crfsw) for JJA. The resolution-induced anomaly differences of cf a) are calculated as the difference of experiment $LR_{MH-PI}$ and $HR_{MH-PI}$. The same procedure is applied for crfsw b). Units are percentages [%] for a) and Watts per square meter [$W/m^2$] for b). Stippled lines as in Figure 3.



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
