# Peer review of "Impact of model resolution on Holocene climate simulations of the Northern Hemisphere"

_Geoscientific Model Development, 2018_

## Referee Comment (RC1) · Anonymous Referee #1 · 15 Oct 2018

Review of "Impact of model resolution on Holocene climate simulation of the Northern Hemisphere" by Axel Wagner, Gerrit Lohmann, Mattias Prange.

This work analyses the impact of resolution on Holocene Northern Hemisphere temperature and atmospheric circulation in ECHAM5 model. Authors found that winter temperature differences among sensitivity experiments are mainly due to changes in the orography and resolution, both affecting the pattern of stationary waves and transient eddies. On the other hand, summer temperature differences are attributable to difference in the cloud cover due to different subgrid parametrization between low and high resolution sensitivities.

General comments: The paper is overall well written and addresses very well the problem of the resolution impact on past temperature with a dedicated set of sensitivity

experiments well designed. I think however that in order to be published, further investigation is needed, especially on the dynamical influence on DJF temperatures.

Specific comments: Ln 111: why PMIP2 boundary conditions and not PMIP3?

Ln 306-311: Adding another section specifically on how stationary waves influenced temperature patterns in the past would be a valuable contribution for understanding regional discrepancies between simulation and proxy reconstructions. Furthermore, the discussion section is too long and sometimes seems just a list of previous work findings (seems an extension of the introduction): in my opinion, it can be shorten, focusing only on the discussion of the results.

Ln 314: Which differences are you talking about? You must be more precise. Adding a comparison with proxy reconstructions would be very useful. Reporting previous findings on the discrepancies itself in the discussions is not enough in order to "solve the problem" in this specific contest. You should be able to quantify those discrepancies between simulations and proxy reconstructions and being able to address to which extent, increasing the resolution would help reducing them. In fact, many other reasons can be imputable to the disagreement between simulations and reconstructions: e.g. dust concentration, vegetation cover. . . not only clouds. . .

Technical comments: Ln 113: "present-day" . . . in order to avoid ambiguities you should write "pre-industrial". Ln 114: You should specify LR_oro before - around Ln 108 - 113. Ln 153: ". . . betwee n" . . . is the space a typo? Ln 220: Erase "The" at the beginning of the sentence.

---

## Short Comment (SC1) · 22 Oct 2018

Dear authors,

please note, that if only one model is concerned, the title of a GMD manuscript should state the model name (or its acronym) and a version number. These are also important to know in the case of an evaluation, as different versions might perform differently for the same evaluation procedure. Therefore please change the title of your manuscript accordingly upon revision; e.g., "Impact of model resolution on Holocene climate simulations of the Northern Hemisphere, a case study with ECHAM5.x.y" .

Please be precise about the ECHAM version (also in the main body of the paper). Did you really use ECHAM5.0? The subversion numbers are important as the subversions

differ considerably.

In addition please be more precise about the license conditions of the model. State explicitly how a user can acquire a license.

Best regards,

Astrid Kerkweg (executive Editor)

———————————————————

---

## Referee Comment (RC2) · Anonymous Referee #2 · 25 Oct 2018

This is an interesting study that looks at the effect of model resolution in ECHAM5 on simulated mid-Holocene surface air temperatures. It is clear to follow and quite succinct. I did, however, find it overly vague and sweeping in places. The phrase 'beyond the scope of the present paper/study' is used twice, but in truth I found it hard to actually understand what the scope, or even direct purpose of the present study was. I do believe it needs to go further than showing/describing the simulated surface air temperature anomalies with vague speculations as to the cause, and the two points identified by the authors (a 'detailed model-data comparison' and 'systematic analysis of the atmospheric dynamics') are both good ways to achieve this (I think one or the other might be adequate, both would be better). In general, the current manuscript is vague and often reads like a series of disjointed statements, especially the introduction

and discussion sections. The reader needs to work quite hard to join the dots. Some of the discussion seems quite abstract without explicitly explaining how it relates to the results. The introduction is similarly presented and the result feels like a list of bullet points rather than a flowing explanation. For example, line 228-230: how does this relate to the results presented here? Little context is given -> why study the mid-Holocene at all? What specific problems is it hoped that the authors will solve with this study? Are those problems solved? Is it just a validation of the extensive [cited] list of previous studies that found increasing atmosphere model resolution improves simulations? Are there only improvements demonstrated in the existing literature? Is it simply an exploration of 'what happens if?', but then again I come back to the question of why do this for the mid-Holocene and not simply repeat it for the pre-industrial? I am left with quite a few similar questions, which makes me think I really did miss the point of this study, though I looked for it. More specific comments follow, but in short, while I find the potential behind this manuscript interesting, I am left wondering really what was the point of it? I think there is merit here, and therefore it is just a case of rewriting the manuscript and presenting more analysis with these questions in mind. The differences plotted in the figures are certainly clear, they just need more context in a better defined investigation.

Specific comments:

- The abstract is rather vague, why not contextualise some of the conclusions – give numbers? E.g. does the effect of resolution match the change from mid-Holocene to pre-industrial because the change is small, or because the effect of resolution is big?

- The abstract should also explain why it is interesting to do this for the mid-Holocene.

- The intro/discussion are too disjointed (see comments above), the information in previous studies has not been properly synthesised or presented with clear relevance to this study. I really ma left with a very long reading list to understand the present manuscript and therefore the background and discussion sections are incomplete (the

manuscript needs to work relatively stand-alone).

- Several acronyms appear to be undefined before they are used.

- Figure order is wrong: in the text they appear 2, 1, 3

- There is almost no information about the mid-Holcoene in the introduction; why study it? what outstanding questions are you hoping to answer?

- The results are structured oddly. I suggest restructuring so that the chain of events (from perturbation to result -> i.e. change in T2m) are tracked through: X causes Y resulting in Z (and however many other steps in between). This would be a clearer way to present the underlying mechanisms (not explored enough in the current manuscript version) and how the change in model resolution results in significantly different surface air temperatures. For example, section 3.4 could come first in the results section.

- Line 27-29: this is very vague. The whole paragraph is sweeping, but this sentence is particularly vague.

- Line 51-52: vague – needs explaining.

- Line 55: 'intermodal differences' -> such as? What is the point being made here, I suggest focussing on these and drawing them out, using the previous studies to construct your argument/narrative so that the reader can follow the picture outlined by these previous studies and how your work builds on them.

- Line 63: 'nonlinear processes of medium scales', what are these? This whole paragraph is also too vague and abrupt

- Line 65: byt this point the list of studies and their key findings (vaguely and briefly presented) is becoming repetitive. Could you group this work, synthesise and summarise it, and then use the citations as examples/to illustrate the points being made, rather than listing the findings for each individual study? The list-format is difficult to read and for me to keep focussed on.

- Line 77: 'Here, we. . .dependent results'. What does this sentence actually mean?

- Line 81: 'Low model versions'. What does this mean, low resolution?

- Line 83: 'large discrepancy'. Quantify

- Line 84-85: 'more realistic precipitation patterns' -> very vague. In what sense? More information needed.

- Line 91: add PMIP4 reference: Otto-Bliesner, B.L., Braconnot, P., Harrison, S.P., Lunt, D.J., Abe-Ouchi, A., Albani, S., Bartlein, P.J., Capron, E., Carlson, A.E., Dutton, A., Fischer, H., Goelzer, H., Govin, A., Haywood, A., Joos, F., LeGrande, A.N., Lipscomb, W.H., Lohmann, G., Mahowald, N., Nehrbass-Ahles, C., Pausata, F.S.R., Peterschmitt, J.-Y., Phipps, S.J., Renssen, H., Zhang, Q., 2017. The PMIP4 contribution to CMIP6 – Part 2: Two interglacials, scientific objective and experimental design for Holocene and Last Interglacial simulations. Geoscientific Model Development 10, 3979–4003. https://doi.org/10.5194/gmd-10-3979-2017

- Line 96-98: I don't think you mean 'Chapter's, maybe 'sections'

- Line 111: why not PMIP4?

- Line 115-116: 'i. e. 100 orbital years', actually I do not understand why 10 model years correspond to 100 orbital years

- Line 122: 'realistic variability in SST and SI model forcing fields', according to what?

- Line 151-152: far too vague, explain precisely what causes the changes

- Line 154-156: this is not convincing. Can you investigate and explain the mechanism rather than simply assume, based on spatial correlation. How do you know that the T change is not driving the cloud cover change? I'm not saying you are wrong, I'm just saying that there is not enough evidence for me to see this is right.

- Line 166: '. . .leads to less pronounced. . .' how and why?

- Line 194-195: why does this amplification happen?

- Line 199: '. . .same order of magnitude. . .' (this is repeated quite a few times and I actually got this point the first time, so please remove at least some of the subsequent repetitions). Why is this important? What does it tell us?

- Line 200: 'changes in atmospheric circulation', what changes?

- Line 235-236: this is hard to follow. What is the significance of this statement?

- Line 236-238: so what if they discuss it? Specifically what was learned and how/why is it relevant here? Also seems like a new discussion point that is separate from the previous sentences in the paragraph (new paragraph needed?)

- Line 239: 'advantage' – what advantage?

- Line 246-256: this information should be in the introduction, not first mentioned here

- Line 246: in what way are they in line?

- Line 258: 'influenced' how?

- Line 261: '. . .treatment of the processes on different scales': this is very vague

- Line 264-266: this is also very vague!

- Line 266-271: how does all of this relate to your findings?

- Line 279-280, so what does explain the differences? Is it the changes in stationary wave patterns mentioned in the next sentence? The link is not made clear.

- Line 282-290: can this comparison be plotted? i.e. the 'proxy' data and your model results.

- Line 292: 'Compared to. . .' what is compared to the proxy data? Also, most of this paragraph (info on climate proxies) should at least be in the introduction

- Line 301: 'However. . .' but there is almost no comparison to the data. Why look at the

mid-Holocene at all? What is learned that couldn't be learned form the pre-industrial, for example.

- Line 305-306: in ECHAM5, or can you relate your results to other models (not done in the present manuscript)?

- Line 310 'in a similar way as described in other contexts' -> far too vague

- Line 311-313: this is not convincing from the present manuscript

- Line 314-315: Can you demonstrate this, then? e.g. show similar signals in anomalies between PMIP models? That is an important thing to show if you want to make this point. Otherwise, it is not really implied at all.

- Line 318-320: it is inannpropriate to suddenly introduce atmospheric blocking for the first time at this very late stage in the manuscript, it should be discussed (and in the intro, and in your results) earlier.

- Line 323-325: 'For palaemodel intermcomparison studies. . .circulation models': this can be removed since it is already said in the previous sentence. In general the conclusions are weak and possibly need rewriting once the purpose of the study has really been made clear in the preceding text to answer some of the questions outlined/address that purpose.

---

## Author Comment (AC1) · 29 Nov 2018

Review of "Impact of model resolution on Holocene climate simulation of the Northern Hemisphere" by Axel Wagner, Gerrit Lohmann, Matthias Prange.

This work analyses the impact of resolution on Holocene Northern Hemisphere temperature and atmospheric circulation in ECHAM5 model. Authors found that winter temperature differences among sensitivity experiments are mainly due to changes in the orography and resolution, both affecting the pattern of stationary waves and transient eddies. On the other hand, summer temperature differences are attributable to difference in the cloud cover due to different subgrid parametrization between low and high resolution sensitivities.

[Figure]

General comments: The paper is overall well written and addresses very well the problem of the resolution impact on past temperature with a dedicated set of sensitivity experiments well designed. I think however that in order to be published, further investigation is needed, especially on the dynamical influence on DJF temperatures.

AW: As recommended by the reviewer, a discussion on the dynamical influence on DJF temperatures, particularly stationary wave patterns, has been integrated in the introduction, results and discussion section of the updated version of the manuscript.

Specific comments: Ln 111: why PMIP2 boundary conditions and not PMIP3? AW: Model simulations have been performed with PMIP2 boundary conditions. PMIP3 boundary conditions are equal to PMIP2. Thus, as recommended by the reviewer, we will substitute the PMIP2 reference by PMIP3. See also table PMIP2 – 4 comparison of experimental design. PMIP4 boundary conditions of the pre-industrial period show small changes (see table). For the mid-Holocene, $CO_2$ and $N_2O$ are comparable to PMIP2 and PMIP3. According to PMIP4, our $CH_4$ estimation is overestimated by 8,9 %. References: https://pmip2.lsce.ipsl.fr, https://pmip3.lsce.ipsl.fr, https://pmip4.lsce.ipsl.fr/doku.php

Action: We will insert a sentence about PMIP4 into the new version of the manuscript.

Ln 306-311: Adding another section specifically on how stationary waves influenced temperature patterns in the past would be a valuable contribution for understanding regional discrepancies between simulation and proxy reconstructions. Furthermore, the discussion section is too long and sometimes seems just a list of previous work findings (seems an extension of the introduction): in my opinion, it can be shorten, focusing only on the discussion of the results.

AW: The discussion section has been shortened, reorganized, more focus on the discussion of own results, and a paragraph about stationary wave patterns and atmospheric blocking has been added in the new version. Parts of the discussion section has been moved to the introduction. Changed will be presented in the revised

manuscript.

Ln 314: Which differences are you talking about? You must be more precise. Adding a comparison with proxy reconstructions would be very useful. Reporting previous findings on the discrepancies itself in the discussions is not enough in order to "solve the problem" in this specific contest. You should be able to quantify those discrepancies between simulations and proxy reconstructions and being able to address to which extent, increasing the resolution would help reducing them. In fact, many other reasons can be imputable to the disagreement between simulations and reconstructions: e. g. dust concentration, vegetation cover ... not only clouds ...

AW: The discussion section has been extended by a paragraph about proxy-model comparisons. Furthermore, large-scale improvements/deteriorations of high resolution simulations are discussed in the framework of proxy-results. Changed will be presented in the revised manuscript.

Technical comments: Ln 113: "present-day" ... in order to avoid ambiguities you should write "pre-industrial". AW: Changed. Ln 114: You should specify LR_oro before - around AW: Changed. Ln 108 - 113. Ln 153: " ... between" ... is the space a typo? AW: Changed. Ln 220: Erase "The" at the beginning of the sentence. AW: Changed.

---

## Author Comment (AC2) · 29 Nov 2018

General comments I did, however, find it overly vague and sweeping in places. The phrase 'beyond the scope of the present paper/study' is used twice, but in truth I found it hard to actually understand what the scope, or even direct purpose of the present study was. I do believe it needs to go further than showing/describing the simulated surface air temperature anomalies with vague speculations as to the cause, and the two points identified by the authors (a 'detailed model-data comparison' and 'systematic analysis of the atmospheric dynamics') are both good ways to achieve this (I think one or the other might be adequate, both would be better). In general, the current manuscript is vague and often reads like a series of disjointed statements, especially the introduction and discussion sections. The reader needs to work quite hard to join the dots. Some

of the discussion seems quite abstract without explicitly explaining how it relates to the results. The introduction is similarly presented and the result feels like a list of bullet points rather than a flowing explanation. For example, line 228-230: how does this relate to the results presented here? Little context is given -> why study the mid-Holocene at all? What specific problems is it hoped that the authors will solve with this study? Are those problems solved? Is it just a validation of the extensive [cited] list of previous studies that found increasing atmosphere model resolution improves simulations? Are there only improvements demonstrated in the existing literature? Is it simply an exploration of 'what happens if?', but then again I come back to the question of why do this for the mid-Holocene and not simply repeat it for the pre-industrial? I am left with quite a few similar questions, which makes me think I really did miss the point of this study, though I looked for it. More specific comments follow, but in short, while I find the potential behind this manuscript interesting, I am left wondering really what was the point of it? I think there is merit here, and therefore it is just a case of rewriting the manuscript and presenting more analysis with these questions in mind. The differences plotted in the figures are certainly clear, they just need more context in a better defined investigation.

AW: Abstract and introductory section have been reorganized to specify the overly vague statements. In addition both sections have been expanded by

a) a broader context with a focus on the mid-Holocene and pre-industrial period,

b) a more detailed description about the scientific problems that are being addressed in this study,

c) a detailed discussion about the difficulty of model-proxy comparisons.

To a) the mid-Holocene to pre-industrial period is an excellent test case for climate models under modern climate conditions. Furthermore, the mid Holocene is one of the most studied paleo time slices with respect to both data and models (e.g. PMIPs; nearly modern ice sheets and sea level) with a relatively small "signal-to-noise ratio".

The available variety of proxy datasets allows the validation of climate simulations.

To b) the key finding of this study: sensitivity experiments of climate models with the only focus on model grid resolution provide significantly different results. The order of magnitude of these results are - on a regional geographic scale - as high as the proxy-reconstructed mid-Holocene to pre-industrial climate trend itself. What causes this significant difference of resolution-dependent model results? Initial and boundary conditions remain unchanged, also the parameterization schemes of ECHAM 5.3. Exceptions of this rule are 1) the damping time of the highest resolvable wavenumber in the horizontal diffusion scheme, 2) the subgrid-scale parameters in the orographic drag scheme, 3) the adjustment time scale utilized in the penetrative convection parameterization and, 4) the radiation budget terms. Isolating the effects of cloud distributions, cloud radiative forcing and orographic drag scheme demonstrate the triggers of our findings. Orographic induced effects are more prominent during boreal winter. In the connection to orographic resolution, the stationary wave pattern in the atmosphere and atmospheric blocking shows significant changes. On the other hand, boreal summer differences of model-resolution are primarily triggered by radiation parameterizations.

To c) The findings of this study demonstrate the challenges of model-proxy comparisons, in particular proxy studies referencing model studies of different resolutions. A solution to this issue could be the comparison of multi-model approaches with proxy-reconstructions. However, in this context multi-model comparisons should be interpreted with special care as, like the findings of this study show, they can extinguish climate trends due to the magnitude of model biases.

The results section has been reorganized: dismantling the bullet-point structure to a more flowing pattern.

The discussion section has been shortened and reorganized in the new version of the manuscript. More focus has been drawn on discussing own results. Thus, two paragraphs have been added to the discussion section: 1) A detailed comparison of model

results and proxy reconstructions as well as a discussion about stationary waves, atmospheric blocking and their connection to our findings.

Specific comments: - The abstract is rather vague, why not contextualize some of the conclusions – give numbers? E.g. does the effect of resolution match the change from mid-Holocene to pre-industrial because the change is small, or because the effect of resolution is big?

AW: Model sensitivity studies have been performed with the general circulation model ECHAM5.3. The runs carried out, transient time-slices, include low (T31L19) and high (T106L31) resolution simulations for the mid-Holocene (MH) and pre-industrial (PI) period. The periods are a good test case for modern climate conditions, the MH provide high quality proxy reconstructions and a well-defined model setup with realistic boundary conditions for paleoclimate simulations. The physical packages of T31L19 and T106L31 remain identical while parameterizations like the subgrid-scale parameters in the orographic drag scheme or radiation budget terms differ. Thus, the main focus is drawn on orographic and radiation effects. Starting from the time-slices, two types of model anomalies are defined: first, T31L19 anomalies of MH minus PI and second, T106L31 anomalies of MH and PI. This results in MH to PI climate trends, low and high resolution ones. The model trends are compared with each other and with proxy reconstructions. Key findings are resolution-dependent model biases are of the same magnitude as the proxy-reconstructed climate trend (mid-Holocene to pre-industrial climate). This because the climate change from mid-Holocene to PI is relatively small (see above) and because the resolution effect is relatively large. Action: We will clarify this point in our revised version.

- The abstract should also explain why it is interesting to do this for the mid-Holocene.

AW: See reply to the first specific comment. "The mid-Holocene to pre-industrial period is a well-defined test case of current climate models . . ."

- The intro/discussion are too disjointed (see comments above), the information in pre-

vious studies has not been properly synthesized or presented with clear relevance to this study. I really ma left with a very long reading list to understand the present manuscript and therefore the background and discussion sections are incomplete (the manuscript needs to work relatively stand-alone).

AW: As recommended by the reviewer introduction and discussion section will be properly synthesized in the revised manuscript. Also the demanded background information will be provided and the discussion section reorganized, partly shortened, partly expanded by a paragraph about stationary waves and atmospheric blocking.

- Several acronyms appear to be undefined before they are used.

AW: We will go through all the acronyms.

- Figure order is wrong: in the text they appear 2, 1, 3

AW: The figure order in Line 119 will be changed in the revised manuscript.

- There is almost no information about the mid-Holocene in the introduction; why study it? what outstanding questions are you hoping to answer?

AW: See reply to the first specific comment.

- The results are structured oddly. I suggest restructuring so that the chain of events (from perturbation to result -> i.e. change in T2m) are tracked through: X causes Y resulting in Z (and however many other steps in between). This would be a clearer way to present the underlying mechanisms (not explored enough in the current manuscript version) and how the change in model resolution results in significantly different surface air temperatures. For example, section 3.4 could come first in the results section.

AW: The results section is structured as followed: modeled temperature differences during boreal winter, followed by temperature differences during boreal summer, followed by a subsection about the isolated effect of orography and closes with the combined effect of orography and resolution.

- Line 27-29: this is very vague. The whole paragraph is sweeping, but this sentence is particularly vague.

AW: Paragraph restructured in the revised manuscript with a broader introduction to the topic.

- Line 51-52: vague – needs explaining.

AW: These resolution-dependent differences of these variables lead to changes in surface temperature (Roeckner et al., 2006, Dallmeyer, 2008). The context of the statement is provided between lines 48 and 51.

- Line 55: 'intermodal differences' -> such as? What the point is being made here, I suggest focusing on these and drawing them out, using the previous studies to construct your argument/narrative so that the reader can follow the picture outlined by these previous studies and how your work builds on them.

AW: Added further information about the intermodal differences. The paragraph focuses on research studies about model simulations of a specific model. Boundary and initial conditions remain unchanged. Only the models grid resolution and thus internal parameterizations are adjusted. The results show that changes in the model resolution regionally exceed trends. The same effect is apparent in our model study of the Holocene. A comparison of model results and proxies can lead to misleading conclusions.

- Line 63: 'nonlinear processes of medium scales', what are these? This whole paragraph is also too vague and abrupt

AW: . . . nonlinear processes of medium scales like for example eddy transition, convective precipitation, cloud forcing and distribution. The paragraph will be rephrased in the revised manuscript.

- Line 65: by this point the list of studies and their key findings (vaguely and briefly presented) is becoming repetitive. Could you group this work, synthesize and summarize

it, and then use the citations as examples/to illustrate the points being made, rather than listing the findings for each individual study? The list-format is difficult to read and for me to keep focused on.

AW: As suggested, outcomes have been grouped and synthesized.

- Line 77: 'Here, we ... dependent results'. What does this sentence actually mean?

AW: The here presented research study can be regarded as an additional research study of purely resolution-dependent studies of the Holocene period.

- Line 81: 'Low model versions'. What does this mean, low resolution?

AW: Dong and Valdes, 2000, apply the spectral model, the U.K. Universities Global Atmospheric Modeling Programme (UGAMP) general circulation model. The model solves meteorological fields by trigonometric functions. The low-resolution model experiments show a horizontal spectral truncation at total wavenumber 21. Action: We clarify this in the manuscript.

- Line 83: 'large discrepancy'. Quantify

AW: Integrated a quantification of the outcome of Jost et al. 2005 within the revised version of the manuscript.

- Line 84-85: 'more realistic precipitation patterns' -> very vague. In what sense? More information needed.

AW: Pollen data indicates drier conditions compared to model results. High-resolution model results show trends in precipitation that are more realistic compared to low resolution simulations. In the revised version, it will be specified.

- Line 91: add PMIP4 reference: Otto-Bliesner, B.L., Braconnot, P., Harrison, S.P., Lunt, D.J., Abe-Ouchi, A., Albani, S., Bartlein, P.J., Capron, E., Carlson, A.E., Dutton, A., Fischer, H., Goelzer, H., Govin, A., Haywood, A., Joos, F., LeGrande, A.N., Lipscomb, W.H., Lohmann, G., Mahowald, N., Nehrbass-Ahles, C., Pausata, F.S.R., Peterschmitt,

[Figure]

J.-Y., Phipps, S.J., Renssen, H., Zhang, Q., 2017. The PMIP4 contribution to CMIP6 – Part 2: Two interglacials, scientific objective and experimental design for Holocene and Last Interglacial simulations. Geoscientific Model Development 10, 3979–4003. https://doi.org/10.5194/gmd-10-3979-2017

AW: Thanks. Added.

- Line 96-98: I don't think you mean 'Chapter's, maybe 'sections'

AW: Replaced chapter by section.

- Line 111: why not PMIP4?

AW: Model simulations have been performed with PMIP2 boundary conditions. PMIP3 boundary conditions are equal to PMIP2. Thus, as recommended by the reviewer, we will substitute the PMIP2 reference by PMIP3. See also table PMIP2 – 4 comparison of experimental design. PMIP4 boundary conditions of the pre-industrial period show small changes (see table). For the mid-Holocene, $CO_2$ and $N_2O$ are comparable to PMIP2 and PMIP3. According to PMIP4, our $CH_4$ estimation is overestimated by 8,9 %. References: https://pmip2.lsce.ipsl.fr, https://pmip3.lsce.ipsl.fr, https://pmip4.lsce.ipsl.fr/doku.php

Action: We mention the different PMIP versions in the revised manuscript.

- Line 115-116: 'i. e. 100 orbital years', actually I do not understand why 10 model years correspond to 100 orbital years

AW: In our set up, mid-Holocene is computed with a transient astronomical forcing 6400 to 6000 before present (BP), whereas PI from 1400 to 1800 Common Era (CE). An acceleration technique was employed in which the orbital forcing is accelerated by a factor of 10 (Lorenz and Lohmann, 2004). Lorenz, S. J., and Lohmann, G.: Acceleration technique for Milankovitch type forcing in a coupled atmosphere-ocean circulation model: method and application for the Holocene, Climate Dynamics, 23, 7-8, 727-743, 2004. The effect of using this technique or the typical time slice is minor.

- Line 122: 'realistic variability in SST and SI model forcing fields', according to what?

AW: Added further information. The variability addresses the conditions during the mid-Holocene and pre-industrial time-slice simulations.

- Line 151-152: far too vague, explain precisely what causes the changes

AW: Focusing on Europe, high-resolution climate simulations reproduce trends in precipitation more closely compared to their low-resolution counterparties. However, the simulated trends of the higher resolved runs are still more humid as depicted by proxy-reconstructions (Jost et al., 2005). We refer to previous studies that have shown the positive correlation between cloud cover and surface cloud radiative forcing during summer at high latitudes (e.g. Vavrus et al., 2009).

Curry, J.A., J.L. Schramm, W.B. Rossow, and D. Randall, 1996: Overview of Arctic Cloud and Radiation Characteristics. J. Climate, 9, 1731–1764, https://doi.org/10.1175/1520-0442(1996)009<1731:OOACAR>2.0.CO;2

- Line 154-156: this is not convincing. Can you investigate and explain the mechanism rather than simply assume, based on spatial correlation. How do you know that the T change is not driving the cloud cover change? I'm not saying you are wrong, I'm just saying that there is not enough evidence for me to see this is right.

AW: Subtracting the mean surface pressure fields of T31MH-PI by and T106MH-PI results in a high pressure anomaly over parts of Eurasia (not shown). As a consequence, the air masses subside, dry out due to the adiabatic heating and cause a reduced total cloud cover. This in turn effects then surface shortwave cloud radiation forcing that in turn changes surface temperatures. We further refer to previous studies demonstrating the effect of cloud cover on the surface radiative budget in summer (e.g. Vavrus et al., 2009).

- Line 166: '... leads to less pronounced ...' how and why?

AW: Two model simulations with identical model boundary and initial conditions have

been performed. Only the orographic mask of the model has been adjusted. Thus, the only input parameter that can lead to changes of the described effects is orographic induced. Internal interactions that are caused by the adjusted orography couldn't be isolated and are not known to us.

- Line 194-195: why does this amplification happen?

AW: Following the explanation of Line 166, the winter case amplification is partly a consequence of changed pressure distributions (Fig. 4).

- Line 199: '... same order of magnitude ...' (this is repeated quite a few times and I actually got this point the first time, so please remove at least some of the subsequent repetitions). Why is this important? What does it tell us?

AW: The consequence of this finding is quite important on a regional scale, temperature anomalies of the mid-Holocene to pre-industrial are of the same magnitude as model resolution errors. Thus, a comparison of model and proxy data sets for the Holocene can lead to quite divergent conclusions.

- Line 200: 'changes in atmospheric circulation', what changes?

AW: The explanation is given in Line 200 – 208, and is in the new version.

- Line 235-236: this is hard to follow. What is the significance of this statement?

AW: Model bias is larger than temperature anomalies. The statement holds for the temperature difference between mid-Holocene and pre-industrial. The model-error has the potential to reserve the temperature evolution. And as a consequence, proxy-model comparisons are highly dependent on the model used.

- Line 236-238: so what if they discuss it? Specifically what was learned and how/why is it relevant here? Also seems like a new discussion point that is separate from the previous sentences in the paragraph (new paragraph needed?)

AW: Separated the discussion point in a new paragraph. The Holocene cooling trend
of summer surface temperatures over continental Eurasia and North America can be linked to PMIP2 boundary conditions, in particular the tropical sea surface temperatures. They affect extratropical pressure systems and the flow of air masses. The advection of air masses can influence the temperature trend.

- Line 239: 'advantage' – what advantage?

AW: Fixing SSTs isolates the atmospheric processes from oceanic processes. Coupling to an ocean would add complexity to the problem, making the interpretation of results even more difficult.

- Line 246-256: this information should be in the introduction, not first mentioned here

AW: Adjusted.

- Line 246: in what way are they in line?

AW: The results are in line with the findings of the quoted publications.

- Line 258: 'influenced' how?

AW: A detailed description of the influence has been added.

- Line 261: '... treatment of the processes on different scales': this is very vague

AW: Cancelled.

- Line 264-266: this is also very vague!

AW: Statement removed in the revised manuscript.

- Line 266-271: how does all of this relate to your findings?

AW: A more explicit relation to our findings is presented in the revised version of the manuscript.

- Line 279-280, so what does explain the differences? Is it the changes in stationary wave patterns mentioned in the next sentence? The link is not made clear.

AW: Please explain in detail.

- Line 282-290: can this comparison be plotted? i.e. the 'proxy' data and your model results.

AW: Yes.

- Line 292: 'Compared to ...' what is compared to the proxy data? Also, most of this paragraph (info on climate proxies) should at least be in the introduction

AW: The here presented research study focuses on model simulations. The introduction provides intra- and intermodal studies of the present and past climate states. A paragraph about climate proxies would cover the intended track of this study.

- Line 301: 'However ...' but there is almost no comparison to the data. Why look at the mid-Holocene at all? What is learned that couldn't be learned from the pre-industrial, for example?

AW: A motivational part about the importance of the mid-Holocene has been added in the introduction as well as the abstract (revised manuscript).

- Line 305-306: in ECHAM5, or can you relate your results to other models (not done in the present manuscript)?

AW: Results can be related to other models like CCSM which can be the starting point for a systematic analysis in PMIP. We mention this in the revised version.

- Line 310 'in a similar way as described in other contexts' -> far too vague

AW: As recommended by the reviewer, a paragraph about stationary waves and atmospheric blocking has been added to the introduction and discussion section. The cited line 310 is an integral part of this paragraph and will be reformulated in the revised manuscript.

- Line 311-313: this is not convincing from the present manuscript

AW: Answer provided for statement Line 151-153.

- Line 314-315: Can you demonstrate this, then? e.g. show similar signals in anomalies between PMIP models? That is an important thing to show if you want to make this point. Otherwise, it is not really implied at all.

AW: The setup of our model simulations demonstrates the statement of line 314-315. A comparison of PMIP2, PMIP3 and PMIP4 models can be conducted. However, the focus is not a comparison of PMIP models, rather the presentation of the ECHAM 5.3 results.

- Line 318-320: it is inappropriate to suddenly introduce atmospheric blocking for the first time at this very late stage in the manuscript, it should be discussed (and in the intro, and in your results) earlier.

AW: Fair enough statement. Introduced atmospheric blocking at an earlier stage.

- Line 323-325: 'For palaemodel intercomparison studies ... circulation models': this can be removed since it is already said in the previous sentence. In general the conclusions are weak and possibly need rewriting once the purpose of the study has really been made clear in the preceding text to answer some of the questions outlined/address that purpose

AW: Removed.

---

## Author Comment (AC3) · 29 Nov 2018

Dear authors, please note, that if only one model is concerned, the title of a GMD manuscript should state the model name (or its acronym) and a version number. These are also important to know in the case of an evaluation, as different versions might perform differently for the same evaluation procedure. Therefore please change the title of your manuscript accordingly upon revision; e.g., "Impact of model resolution on Holocene climate simulations of the Northern Hemisphere, a case study with ECHAM 5.x.y". Please be precise about the ECHAM version (also in the main body of the paper). Did you really use ECHAM 5.0? The subversion numbers are important as the subversions differ considerably. In addition please be more precise about the license conditions of the model. State explicitly how a user can acquire a license.

[Figure]

AW: Adjusted the paper's title as recommended by A. Kerkweg. Complemented the model's subversion in title and main body. Added the contact link (http://mpimet.mpg.de/en/science/models/license) as entry point of the acquiring process of getting an ECHAM 5.3 license.

—————————————